# Anxiety as a mediator of relationships between perceptions of the threat of COVID-19 and coping behaviors during the onset of the pandemic in Poland

**Marzena Cypryańska**[1]*, **John B. Nezlek**[2,3]

**1** Department of Psychology, SWPS University of Social Sciences and Humanities, Warsaw, Poland,
**2** Institute of Psychology, SWPS University of Social Sciences and Humanities, Poznań, Poland,
**3** Department of Psychological Sciences, College of William and Mary, Williamsburg, Virginia, United States of America

* mcypryanska@swps.edu.pl

## Abstract

At the beginning of the COVID-19 pandemic (10–14 March, 2020) we conducted a survey (n = 1028) of a nationally representative sample (age, sex, and locale) in Poland. Respondents indicated how strong they thought the threat was to themselves, to Poland, and the world. They also described their emotional reactions to the pandemic, which we used to calculate three scores: *Anxiety*, *Hopelessness*, and *Panic*. Respondents also indicated how often they engaged in various coping behaviors and how much they supported different types of economic sacrifice. We used these responses to calculate measures that we labelled as *Spread Prevention* (e.g., social distancing), *Self-preservation* (food stockpiling), and *Economic Sacrifice* (e.g., fighting COVID-19 regardless of the cost). Multiple regression analyses found that perceived threat to self was the most reliable predictor (positive) of emotional reactions and of coping behaviors, and that *Anxiety* was the most reliable predictor (positive) of *Spread prevention* and *Economic sacrifice*. *Panic* predicted (positively) *Self-preservation*. A series of mediation analyses found that *Anxiety* mediated relationships between threat and coping behaviors, and that *Panic* mediated the relationship between perceived threats and *Self-preservation*. In addition, we found that scores on all measures, except *Panic*, increased following the announcement of the first COVID-19 related fatality in Poland, which occurred on the third (middle) day of the study. The mediational relationships we found did not vary as a function of whether the data were collected before or after this announcement. The present results suggest that emotional reactions to perceived threats can serve an instrumental function by providing the motivation to engage in coping behaviors. Such a mechanism complements much research on stress that has focused on how coping mediates threat-emotion relationships.

**Data Availability Statement:** The data underlying the results presented in the study are available from https://osf.io/8mjbr/.

**Funding:** The author(s) received no specific funding for this work.

**Competing interests:** The authors have declared that no competing interests exist.

## Introduction

The world is facing the threat of the coronavirus, which the Secretary General of the UN said required a "War-time plan" [1]. The day the first draft of this paper was submitted (April, 7) more than 1,300,000 people around the world had been infected with COVID-19, and approximately 75,000 people had died due to complications arising from the virus. Moreover, both the number of new cases and deaths has risen over time, and it is not clear when the pandemic will end.

Governments across the world are trying to find ways to stop or at least slow down the spread of the disease, with an important goal of reducing the likelihood that health care systems will be overloaded. The term "flattening the curve" is now in common use to describe the effort to reduce the prevalence of cases, and doing this requires the action of both governments and individuals. Governments can provide treatment facilities, but unless individuals take action, such facilities will certainly be overloaded. For example, to protect themselves as well as to slow the spread of COVID-19 individuals need to do things such as wash their hands regularly, practice social distancing, and use face masks. We note that when the present study was conducted the WHO had not yet recommended the use of face masks as a preventative mechanism. Whatever the methods, the success of the fight against COVID-19 depends, in part, on how thoroughly people follow the recommendations that prevent them from contracting the virus.

The present study examined relationships between people's perceptions of the threat of COVID-19 and their coping behaviors. We defined coping behaviors as what people were doing in response to the perceived threat of COVID, including behaviors that were recommended by the authorities (e.g., social distancing) and those that were not (e.g., hoarding). Moreover, the study examined the role that negative emotional reactions to these threats played in how people coped. NB: We use the term "emotional reactions" to refer to a broader range of constructs than is inherent in formal models of emotions [e.g., 2]. Nevertheless, we believe that this broader use is consistent with how these constructs are discussed in research on pandemics.

Our focus on relationships between threat and coping and between threat and emotional reactions was based on previous research that has consistently found that the perceived threat of an infectious disease is positively related to engaging in coping behaviors [3–5]. As noted by Bavel et al. [6], a summary of the possible applications of social science to understanding the CVOID pandemic, "people are less likely to die from over-reaction than from under-reaction, that is, not responding to signs of danger until it is too late" (p. 462). In other words, although many may think that fear paralyzes people, fear can motivate people to act.

In parallel, research has also found that anxiety is positively related to engaging in coping behaviors. For example, a review of research on reactions to the H1N1 influenza (Swine flu) found support for both of these relationships [4]. See also two studies published after this review, Van Der Weerd, Timmermans, Jma Beaujean, Oudhoff, and van Steenbergen [7] who reported positive relationships between fear/worry and adaptive behaviors in NL, and Liao, Cowling, Lam, and Fielding [8] who found that worry about H1N1 was positively related to adaptive coping in Hong Kong.

More specifically, we examined if negative emotional reactions mediate relationships between perceived threat and coping behavior. Such a possibility is suggested by thinking of emotion as a source of motivation, a conceptualization consistent with the fact that the two words share a Latin root, emovere/movere "to move." Such a possibility was also suggested by Witte in her "Extended Parallel Process Model" ([9,10]). As noted by Witte ([9], p. 331) "Fear is a negatively-valenced emotion, accompanied by a high level of arousal, and *is elicited by a*

*threat* [emphasis added] that is perceived to be significant and personally relevant." Such a proposition is consistent with a mediational model in which fear mediates the relationship between perceived threat and coping behavior.

Such mediational relationships can also be understood in terms of Lazarus and Folkman's model of stress and coping [11]. Within their model, the recognition of threat (primary appraisal) initiates a process that includes emotional reactions, the end result of which is coping behavior. In fact, Folkman and Lazarus [12] noted that: "Historically, coping has been viewed primarily as a response to emotion" (p. 6).

Nevertheless, to our knowledge, no previous study has examined the possibility that emotional reactions mediate relationships between stress and coping within the context of reactions to epidemics. Much of the previous research on stress and coping (within the context of epidemics or not) has emphasized (if not considered exclusively) the possibility that coping mediates relationships between threat and emotional reactions to threat [13]. Although valuable, we believe that this research needs to be complemented by research that considers the possibility that emotional reactions mediate threat-coping relationships.

Such mediational relationships are consistent with Witte's assertion that threats elicit emotions and Folkman's and Lazarus's summary statement about how emotions elicit coping. People perceive a threat, they become aroused, and then they act [11]. This explanation assumes that emotions provide the energy (the motivation) to act. Moreover, we think a threat-emotion-coping sequence is more likely at the beginning of a pandemic, when the threat is being realized and people are deciding how to cope, than a threat-coping-emotion sequence, which may occur later after people have coped and re-evaluated a threat. We consider the issue of causal sequences in the discussion.

In this paper, we focus on the mediating role of negative emotional reactions in relationships between perceived threat and coping behavior in the context of coping with COVID-19. Given the relative lack of attention to such mediational relationships, we considered the possibility that different emotional reactions might mediate relationships between different types of perceived threat and different types of coping behaviors. This included the possibility that negative emotional reactions other than anxiety might mediate relationships between perceived threats and coping. Would any negative emotional reaction mediate relationships between threat and coping, or is the mediation specific to anxiety? Given the lack of research on this specific topic, we examined such relationships on an exploratory basis.

Efficacy, defined as the likelihood that a coping behavior will address the problem posed by a threat, is an important feature of most, if not all, models of reactions to threats and stress ([9,11]). In these models, behaviors that cannot be enacted or are not thought to be effective will not be exhibited. In the present study we measured coping behaviors that were practical (i.e., easily accomplished) and that were recommended by the WHO and government officials (i.e., known to be effective). We also measured coping responses that were not recommended by the WHO and government officials but were responses that were being reported in the press. We return to this issue in the discussion.

We should note that the present study was conducted within the context provided by previous research and theory on pandemics per se, with a focus on fear. Responses to pandemics can be understood from multiple perspectives, and the present study was not designed to address these different perspectives. We thought that a clear and precise focus on one model (Witte's) would provide a comprehensible and accessible account of reactions to the COVID pandemic in terms of the components of her model. We discuss complementary contexts and perspectives at the close of this article.

Finally, as explained below, during the middle of our study, the first Polish fatality due to COVID-19 occurred. We did not anticipate that Poland would experience its first fatality

during our study (although it was a near certainty that people would die at some time), but this
event did provide the opportunity to examine how responses to COVID-19 changed as a result
of the occurrence of this event.

## Method

### Context

The WHO declared the COVID-19 as a pandemic on 11 March, 2020 [14], and the data
described in this study were collected between 10 and 14 March, 2020. The first case of
COVID-19 in Poland was officially announced on 4 March. On 9 March the official report was
that 17 people were infected with COVID-19 in Poland, and by the end of the last day of the
survey 104 people were reported as having been infected with COVID-19. Although the poli-
cies now in effect in Poland had not been put into place, the government had announced that
sporting events and mass meetings of any kind were canceled (10 March), and that schools,
theaters, museums, cinemas, and concert halls were closed (11 March), and finally the govern-
ment declared an epidemic emergency (13 March). Following this, the external borders were
closed on 14 and 15 March 14, initially for 10 days. On 25 March the borders were ordered
closed until at least 13 April.

It is particularly important to note that although social distancing had been recommended
while our study was being conducted, there were no formal restrictions on people's day to day
activities. That is, residents could choose to isolate themselves (e.g., stay at home) or not. As
intended, our data were collected before the full brunt of COVID-19 was felt in Poland but
after there was some awareness of the severity of the problem and some sense of the actions
the government would be taking to deal with the problem.

Conducting our study in Poland provided numerous advantages. First, there is functionally
100% literacy, so we could be certain that respondents would understand public service
announcements regarding COVID-19. Second, the society is very homogeneous in terms of
ethnicity, which probably reduced the influence that scapegoating minorities had on
responses. Third, the country is rated to have an intermediate risk of infectious diseases, which
meant respondents did not have a basis for being unusually optimistic or pessimistic about
how their country would deal with COVD-19. Source for all: CIA Factbook [15].

### Procedure and sample

We contracted with Ariadna Research Panel, PL [16] to collect data from a nationally repre-
sentative sample in Poland. Ariadna is a private research firm that collects data from members
of panels. Participants are compensated by Ariadna with credits that they can use at various
retail outlets. We requested a sample that was representative in terms of age, sex, and place of
residence (e.g., size of city). We collected data from 255 respondents on 10 March, 108 on 11
March, 154 on 12 March, 464 on 13 March, and 173 on 14 March. Moreover, Ariadna
obtained informed consent from participants and ensured compliance with RODO protocols.
The study was approved by: Komisja ds. Etyki Badań Naukowych, Uniwersytet SWPS, Filia w
Poznaniu, Wydział Psychologii i Prawa. Approval number 2020-18-11. Data were collected
anonymously. Participants were free to terminate participation at any time.

The initial sample was 1054. To reduce the influence of careless responding, we eliminated
26 participants whose average response time per item was less than 2sec. The final sample con-
sisted of 549 women and 479 men, with an average age of 44.4yo ($SD$ = 15.8yrs). Forty-four
percent of the sample lived in locales/cities with 20,000 or fewer residents, 22.5% lived in cities
with between 20,000 and 99,000 residents, 19.2% lived in cities between 100,000 and 500,000
residents, and 14.7% lived in cities with more than 500,000 residents. Approximately one-third

(33.6%) had a university degree of some kind, another third (32.9%) had the Polish equivalent of a high school degree, and 14.2% had not completed high school. The raw data described in this article are available at OSF [17]. In order, we measured four constructs: perceived threat, emotional reactions, coping behaviors, and support for COVID prevention policies.

## Measures of emotional reactions

Our measures of emotional reactions were based upon the affective circumplex [18], a model that distinguishes positive and negative emotions and activated and deactivated emotions. Given the nature of COVID-19, we measured negative emotional reactions, and we distinguished activated (e.g., anxiety and fear) and deactivated (e.g., sadness, hopeless) emotional reactions. Activated negative emotional reactions were defined in terms of being anxious, scared, worried, concerned, and fearful. Deactivated negative emotional reactions were defined in terms of being powerless, helpless, hopeless, woebegone, and sad. See [19] for a similar approach to distress related to climate change.

In addition, we measured two emotional reactions that we thought would have specific relevance to reactions to COVID-19, feeling panicked and being paralyzed by fear. Although some may think of fear and panic as points on the same continuum, panic has a stronger maladaptive component than fear [20]. By definition, panic entails an irrational component. For example, in discussing reactions to disasters, Van Bavel et al. [6] explicitly distinguish panic, which is likely to have maladaptive consequences (e.g., panic-buying), from fear, which may serve to motivate people to work cooperatively to meet the challenges posed by a disaster.

Questions about each emotional reaction were preceded by the stem: "When you think about the coronavirus, how much, if at all, do you feel." Participants responded using five-point scales with endpoints labeled "not at all" to "extremely." The original Polish language items and response scales and English translations of these items and scales are contained in the supplemental materials [17].

## Measures of coping

How people coped with the onset of the coronavirus was an important focus of the present study. To capture the range of possible behavioral responses, we measured 11 behaviors. Some behaviors (six) were recommended by the authorities (e.g., the WHO) as ways to reduce the likelihood that a person would contract COVID-19 while also reducing the speed with which the disease would spread, i.e., "flattening the curve" of new cases. Other behaviors represented what people were doing in reaction to the pandemic.

These behaviors are listed in Table 1 (below). All questions were preceded by the stem: "How often, during the last week, did you take any of the following actions to protect yourself against the coronavirus or the consequences of the coronavirus." All response scales ranged from 1 to 6, with endpoints labeled "not at all" and "to the maximum extent possible." The original Polish language items and response scales and English translations of these items and scales are contained in the supplemental materials [17]. As can be seen from the labels for the response scales, participants were asked to indicate how much their behaviors had changed from normal.

## Measures of perceived threat and support of policies involving economic sacrifice

Using a 7-point scale with endpoints labeled "1 = this is not a threat at all" and "7 = maximum, the condition is critical," participants indicated how much of a threat they thought COVID-19 was to Poland, the world, and to themselves as individuals. Using a 7-point scale with

**Table 1. Coping behaviors: Descriptive statistics and factor loadings.**

| Behavior | M | SD | Factor loadings | |
|---|---|---|---|---|
| | | | Spread Prevention | Self-Preservation |
| *Avoid contact with the sick | 3.61 | 1.49 | .86 | |
| *Wash hands | 3.82 | 1.37 | .78 | |
| *Seek information | 3.92 | 1.42 | .71 | |
| *Avoid leaving home | 3.20 | 1.56 | .70 | |
| *Avoid touching eyes and nose | 2.97 | 1.46 | .65 | |
| *Use gel (hand sanitizer) | 2.74 | 1.53 | .45 | -.33 |
| Store cleaning products | 2.31 | 1.31 | | -.80 |
| Wear masks | 1.35 | .99 | | -.69 |
| Store foods | 2.41 | 1.32 | | -.70 |
| Consume diet supplements | 2.36 | 1.37 | | -.55 |
| Pray | 2.01 | 1.24 | | -.45 |

* WHO recommended activities.

Note: Item loadings less than .25 deleted.

endpoints labeled "1 = definitely not" and "7 = definitely yes," participants indicated how much they supported using societal resources to fight the spread of COVID-19. The items were: (1) To stop the spread of coronavirus we should do what is needed even if it means slowing economic growth, (2) To stop the spread of coronavirus we should do what is needed regardless of the cost, and (3) To counteract the spread of coronavirus, we must all act and give up various things, if appropriate.

## Results

### Calculation of scale scores

**Emotional reactions.** As intended, self-reports of how anxious, scared, worried, concerned, and fearful participants felt formed a reliable scale ($M = 2.85$, $SD = 1.05$, $\alpha = .95$), self-reports of how powerless, helpless, hopeless, woebegone, and sad participants felt formed a reliable scale ($M = 2.66$, $SD = 1.05$, $\alpha = .92$), and self-reports of feeling panicked and paralyzed by fear also formed a reliable scale ($M = 2.28$, $SD = 1.16$, $\alpha = .91$). We labeled these scales "Anxiety," "Hopelessness," and "Panic," respectively.

**Coping behaviors.** Given the lack of previous research and theorizing, we were uncertain regarding if/how to combine participants' reports of the behaviors they exhibited in response to the onset of COVID-19. Given the similar focus of some items (e.g., storing food and storing cleaning supplies), we conducted an exploratory factor analysis following guidelines proposed by [21], i.e., a maximum likelihood extraction followed by oblimin rotation. This analysis produced two factors with eigenvalues greater than 1 (5.39, 1.44), and these two factors accounted for 62.1 percent of the total variance. The loadings for the rotated solution are presented in Table 1.

The results of these analyses were quite clear. The first factor consisted of behaviors that were recommended by the WHO (and other civil authorities) to contain the spread of COVID-19. Given the ultimate goal of these behaviors, we labeled this factor as "Spread Prevention." The second factor consisted of behaviors that were primarily concerned with personal well-being, and we labeled this factor "Self-preservation." Interestingly, "using masks" loaded on the second factor but not on the first factor (less than .25). This is consistent with the fact that when the study was being conducted the WHO was not recommending the

routine wearing of masks by individuals who were not infected with COVID-19. The only item that had a loading of greater than .25 on both factors was using gel (hand sanitizers). The factors were negatively correlated (-.57).

We defined *Spread Prevention* as the mean response to the items: Avoid contact with the sick, Wash hands, Seek information, Avoid leaving home, and Avoid touching eyes and nose (scale: $M = 3.51$, $SD = 1.18$, $\alpha = .87$). We defined *Self-preservation* as the mean response to the items: Store cleaning products, Wear masks, Store foods, Consume diet supplements, and Pray (scale: $M = 2.09$, $SD = .96$, $\alpha = .82$). In the interests of conceptual clarity, using gel was not included in either score, although we should note that the results of the analyses reported below did not vary meaningfully as a function of whether using gel was included as part of either or both of these scale scores.

Note that although items loaded negatively on the second factor, the correlation between the two factors was negative. An explanation of how the signs of loadings are determined in factor analysis is well beyond the scope of this paper, but it should suffice to note that the scale scores we calculated were positively correlated (.58). Finally, participants were more likely to exhibit behaviors consistent with WHO guidelines than they were to exhibit behaviors focused on self-preservation ($t(1027) = 45.4$, $p < .001$).

**Economic sacrifice.** As intended, our three measures of economic sacrifice (spend even if leads to slow growth, spend regardless of cost, and give up things) constituted a reliable scale ($M = 5.44$, $SD = 1.25$, $\alpha = .89$), which we labeled *Economic Sacrifice*.

## Regression analyses: Overview

The present study was designed in part to test a mediational model in which emotional reactions were presumed to mediate relationships between perceptions of threat and coping behaviors. Before examining such possibilities, we conducted a series of regression analyses to help determine exactly what combinations of threat, emotional reactions, and behaviors should be examined. For example, if a threat was not significantly related to a behavior, there would be no reason to examine the possible mediation of this relationship by emotional reactions.

We used multiple regression rather than zero-order correlations to select variables for the mediation analyses to take into account relationships among measures of the same category (i.e., emotional reactions and threat). This also reduced the number of analyses we conducted which reduced the "studywise" error rate. We conducted a set of regression analyses for each path in the proposed mediation analyses: threat as a predictor of coping, emotional reactions as a predictor of coping, and threat as a predictor of emotional reactions. These analyses included the *Economic sacrifice* measure as a measure of coping behavior.

## Relationships between threat and reactions to COVID: Coping behaviors and economic sacrifice

First, we regressed our two coping behavior measures and our measure of support for economic sacrifice onto our three measures of threat. For all three analyses, the overall model was significant: *Spread prevention*: $F(3,1024) = 157.4$, $p < .001$, $R^2 = .32$; *Self-preservation*: $F(3,1024) = 38.5$, $p < .001$, $R^2 = .12$; *Economic sacrifice*: $F(3,1024) = 132.7$, $p < .001$, $R^2 = .28$. The estimated standardized coefficients and the results of the significance tests of these coefficients are presented in Table 2. *Spread Prevention* was significantly (positively) related to all three measures of threat. In contrast, only threat to self was significantly related (positively) to *Self-preservation*. *Economic sacrifice* was significantly (and positively) related to all three types of threat (threat to Poland was $p = .06$).

**Table 2. Relationships between threat and reactions to COVID: Coping behaviors and economic sacrifice.**

|  | Source of threat | | | | | |
|  | Self | | Poland | | World | |
| Outcome | β | t | β | t | β | t |
| Spread Prevention | .29 | 6.84*** | .17 | 3.11** | .16 | 3.26** |
| Self-Preservation | .32 | 6.76*** | .10 | 1.57 | -.07 | 1.26 |
| Economic sacrifice | .15 | 3.35** | .11 | 1.93a | .32 | 6.54*** |

Note: Coefficients accompanied by

*** p < .001

** p < .01

a p = .06.

## Relationships between emotional reactions and coping behaviors and economic sacrifice

Next, we regressed our two measures of coping behavior and our measure of support for economic sacrifice onto our three measures of emotional reactions. For all three analyses, the overall model was significant: *Spread prevention*: $F(3,1024) = 188.4$, $p < .001$, $R^2 = .36$; *Self-preservation*: $F(3,1024) = 137.8$, $p < .001$, $R^2 = .29$; and *Economic sacrifice*: $F(3,1024) = 87.3$, $p < .001$, $R^2 = .20$. The estimated standardized coefficients and the results of the significance tests of these coefficients are also presented in Table 3. *Spread Prevention* was significantly (positively) related only to scores on the *Anxiety* factor. *Self-preservation* was significantly (positively) related only to scores on the *Panic* factor (*Anxiety* was $p = .09$). *Economic sacrifice* was significantly (and positively) related to scores on the *Anxiety* factor and was significantly (negatively) related only to scores on the *Panic* factor.

## Relationships between threat and emotional reactions

Finally, we regressed our three measures of emotional reactions onto our three measures of threat. For all three analyses, the overall model was significant: *Anxiety*: $F(3,1024) = 312.6$, $p < .001$, $R^2 = .48$; *Hopelessness*: $F(3,1024) = 161.3$, $p < .001$, $R^2 = .42$; *Panic*: $F(3,1024) = 151.1$, $p < .001$, $R^2 = .33$. The estimated standardized coefficients and the results of the significance tests of these coefficients are presented in Table 4. Scores on the *Anxiety* measure were significantly (positively) related to all three perceived threats. Scores on the *Hopeless* and *Panic* measures were significantly related (positively) to threats to the self and to Poland, but were not significantly related to perceived threat to the world.

**Table 3. Relationships between emotional reactions and coping behaviors and economic sacrifice.**

|  | Anxiety | | Hopelessness | | Panic | |
| Outcome | β | t | β | t | β | t |
| Spread Prevention | .64 | 10.12*** | .01 | < 1 | -.07 | 1.27 |
| Self-Preservation | .12 | 1.72a | -.04 | < 1 | .47 | 8.67*** |
| Economic sacrifice | .65 | 9.28*** | .10 | 1.46 | -.40 | 6.92*** |

Note: Coefficients accompanied by

*** p < .001

a p = .09.

**Table 4. Relationships between threat and emotional reactions.**

| | Source of threat | | | | | |
| | Self | | Poland | | World | |
| Outcome | β | t | β | t | β | t |
|---|---|---|---|---|---|---|
| Anxiety | .47 | 12.73*** | .14 | 2.96** | .14 | 3.25*** |
| Hopeless | .44 | 11.40*** | .20 | 3.99*** | .06 | 1.31 |
| Panic | .46 | 11.15*** | .11 | 2.10* | .03 | < 1 |

Note: Coefficients accompanied by

*** $p < .001$

** $p < .01$

* $p < .05$.

## Mediation analyses

We examined mediation using a series of PROCESS (v 3.3) analyses, Model 4 [22]. For all analyses, we used bootstrapping with 10,000 samples. To provide the clearest description of mediation, we conducted separate analyses for each combination of threat, coping behavior, and emotional reaction suggested by the results of the multiple regression analyses. We report 95% confidence intervals for effects.

First, we examined how emotional reactions might mediate relationships between threat and *Spread prevention*. The multiple regression analyses found that *Spread prevention* was significantly related to all three measures of threat but was significantly related to only *Anxiety*. Accordingly, we ran three analyses, one in which each type of threat was a predictor and *Anxiety* was the mediator.

The results of these analyses were quite clear. *Anxiety* mediated (partially) the relationship between each type of threat and *Spread prevention*. For threat to self, the total effect of .43 was decomposed into an indirect effect of .24 (CI = .20 to .28) and a direct effect of .19 ($p < .001$, CI = .14 to .25). For threat to Poland, the total effect of .50 was decomposed into an indirect effect of .26 (CI = .22 to .31) and a direct effect of .24 ($p < .001$, CI = .18 to .29). For threat to the world, the total effect of .49 was decomposed into an indirect effect of .26 (CI = .22 to .30) and a direct effect of .22 ($p < .001$, CI = .17 to .28).

For *Self-preservation*, threat to self was the only significant predictor, and *Panic* was the only possible mediator. The analyses found that *Panic* mediated (partially) the relationship between threat to self and *Self-preservation*. The total effect of .23 was decomposed into an indirect effect of .19 (CI = .16 to .22) and a direct effect of .04 ($p < .05$, CI = .001 to .08).

Scores on the *Economic sacrifice* measure were significantly related to all three measures of threat. Threat to self was significantly related to *Anxiety* and *Panic*, and *Anxiety* and *Panic* were significantly related to *Economic sacrifice*. Given this, we examined the extent to which *Anxiety* and *Panic* mediated relationships between threat to self and *Economic sacrifice*. One set of analyses examined the mediating role of *Anxiety* with *Panic* as a covariate, and the other set of analyses examined the mediating role of *Panic* with *Anxiety* as a covariate.

The analyses found that only *Anxiety* mediated (partially) the relationship between threat to self and *Economic sacrifice*. For the analysis in which *Anxiety* was the mediator and *Panic* was the covariate, the total effect of .40 was decomposed into a direct effect of .29 ($p < .001$, CI = .22 to .35), and an indirect effect for *Anxiety* of .12 (CI = .09 to .15). In contrast, for the analysis in which *Panic* was the mediator and *Anxiety* was the covariate, the indirect effect for *Panic* was not significant, .003 (CI = -.011 to .017).

Similar to threat to self, threat to Poland was significantly related to the *Anxiety* and *Panic* measures, and so we examined the extent to which *Anxiety* and *Panic* mediated relationships between threat to Poland and *Economic sacrifice*. As before, one set of analyses examined the mediating role of *Anxiety* with *Panic* as a covariate, and the other set of analyses examined the mediating role of *Panic* with *Anxiety* as a covariate.

Similar to the results of the previous analyses, these analyses found that only *Anxiety* mediated (partially) the relationship between threat to Poland and *Economic sacrifice*. For the analysis in which *Anxiety* was the mediator and *Panic* was the covariate, the total effect of .48 was decomposed into a direct effect of .36 ($p < .001$, CI = .30 to .43) and an indirect effect for *Anxiety* of .12 (CI = .09 to .15). For the analysis in which *Panic* was the mediator and *Anxiety* was the covariate, although the indirect effect for *Panic* was significant, it was small, .017 (CI = .0035 to .0314).

Given that threat to the world was significantly related to only *Anxiety*, we examined the extent to which *Anxiety* mediated the relationship between threat to the world and *Economic sacrifice*. The analysis found that *Anxiety* mediated (partially) the relationship between threat to the world and *Economic sacrifice*. The total effect of .52 was decomposed into an indirect effect for *Anxiety* of .10 (CI = .05 to .15), and a direct effect of .43 ($p < .001$, CI = .36 to .49).

## The role of the first fatality in Poland

On 12 March, at approximately 12:00 (CET), the vice president of Poznan, a city in western Poland, announced that Poland had experienced its first fatality, a 57yo woman who had lived in or around Poznan. This announcement was broadcast widely, including state-run media (e.g., www.tvp.info), and the news spread quickly through other media outlets. This tragic event occurred in the middle of our data collection and provided an opportunity to assess the impact of this news on citizens' feelings and beliefs.

The means for each day for the variables we measured are presented in Table 5. This table also contains the results of t-tests that compared the means of responses provided before the announcement (the entire day for 10 and 11 March, and before 12:00 on 12 March) to the means of responses provided after the announcement (13 and 14 March). Given the

**Table 5. Means for each day of study and for responses before and after notification of first fatality.**

| Variable | March | | | | | First fatality | | |
| | 10 | 11 | 12 | 13 | 14 | Before | After | *t*-ratio |
|---|---|---|---|---|---|---|---|---|
| Sample size | 249 | 106 | 64 | 450 | 73 | 419 | 523 | |
| Threat to self | 4.61 | 4.47 | 4.83 | 5.02 | 5.38 | 4.61 | 5.07 | 5.03*** |
| Threat to Poland | 4.99 | 4.85 | 5.08 | 5.40 | 5.71 | 4.97 | 5.44 | 6.00*** |
| Threat to world | 5.29 | 5.17 | 5.41 | 5.65 | 5.89 | 5.27 | 5.68 | 5.21*** |
| Anxiety | 2.71 | 2.67 | 2.79 | 2.89 | 3.16 | 2.71 | 2.93 | 3.22** |
| Hopelessness | 2.53 | 2.51 | 2.71 | 2.69 | 2.92 | 2.55 | 2.72 | 2.51* |
| Panic | 2.18 | 2.14 | 2.42 | 2.29 | 2.47 | 2.21 | 2.31 | 1.39 |
| Spread prevention | 3.16 | 3.13 | 3.33 | 3.74 | 4.03 | 3.18 | 3.78 | 7.90*** |
| Self-preservation | 1.93 | 1.97 | 2.22 | 2.12 | 2.23 | 1.99 | 2.14 | 2.44* |
| Economic sacrifice | 5.16 | 5.10 | 4.96 | 5.66 | 6.06 | 5.12 | 5.72 | 7.48*** |

Note: 12 March includes only responses before 12:00.

*** $p < .001$

** $p < .01$

* $p < .05$.

uncertainty about exactly when participants may have heard about the first fatality on 12 March, the 86 responses made on 12 March after 12:00 are not included in the summary statistics in this table and were not included in the *t*-tests.

The results of the analyses comparing reports of respondents before and after the fatality were quite clear. Following the first fatality, respondents perceived greater threats to the self, to Poland, and to the world then they did before the first fatality. They were also more anxious and hopeless, they engaged in more coping behaviors, and they were more supportive of economic sacrifice to fight COVID-19.

Changes in means such as these suggest that the results of the previous mediation analyses might simply reflect changes across time in the measures we collected rather than relationships among the constructs the measures represented. To address this issue, we ran a series of PROCESS analyses in which we examined if the mediational relationships we found differed as a function of when the data were collected (before or after the first fatality). In these models (model 59 in the PROCESS macro), all paths were modeled as moderated by first fatality (before vs. after), and as before, we used 10,000 bootstrapped samples with a 95% CI.

The results of these analyses did not find that the mediational effects we reported previously were moderated by time of data collection. In all analyses, the 95% confidence interval of estimate of the critical statistic, "Index of moderated mediation (difference between conditional indirect effects)," included 0.

## Discussion

As expected, we found that emotional reactions to COVID-19 mediated relationships between perceptions of threat and coping behaviors. We found that anxiety was the most reliable mediator of the relationships we found between threat and coping. Anxiety mediated relationships between threats to self, Poland, and the world and engaging in WHO recommended behaviors and supporting policies to combat COVID-19 that required economic sacrifice. Feeling panic/ paralyzed by fear also mediated the relationship between threat to self and self-focused behavior (e.g., hoarding). In all cases, the indirect effects were positive.

These results are consistent with Taylors's conclusion that "A moderate level of fear or anxiety can motivate people to cope with health threats, but severe distress can be debilitating" ([23], p. 24). The means for our measures of emotional reactions to COVID-19 (both before and after the first fatality) were all below 3, the midpoint of the scale, which corresponded to a scale point of "moderately." Such levels correspond to what Taylor was discussing as moderate. Respondents' emotions served as a source of energy for coping while not being strong enough to dysregulate or derail their adaptive functioning.

The publication of Lazarus and Folkman's now classic Stress, Appraisal, and Coping [24] changed how researchers and practitioners conceptualized stress and coping. Lazarus and Folkman proposed that stress and reactions to stress were involved in bidirectional relationships with coping, and much of the research that has followed this model has emphasized how coping can mediate the effects of stress. Put simply, if people believe they can cope with a stressor (i.e., self-efficacy) they will react less strongly to the stressor than if they believe they cannot cope with the stressor.

We do not disagree at all with this conclusion. Nevertheless, we believe that in the search for how the feedback loops suggested by Lazarus and Folkman work, researchers have lost sight of the importance of the initial stage of the stress reaction, i.e., the initial emotional reactions people have to stress. As suggested by Witte [9], fear can mobilize people. If people are not afraid of an event, why should they do anything about it? Certainly, after they have done something this fear may be reduced by what they have done (coping mediating relationships

between threat and emotional reactions), but people need to be motivated to do something, and we believe that fear can perform this function.

Since we conducted our study, other studies have found that fear can motivate protective behaviors within the context of the COVID. For example, in a study of a representative sample in the UK, Harper et al. [25] concluded that "Consistently, the only predictor of positive behavior change (e.g., social distancing, improved hand hygiene) was fear of COVID-19." Similarly, Winter et al. [26] concluded: ". . . there was a significant relationship between FCV-19S scores [a measure of fear] and adherence to the lockdown rules that were implemented in New Zealand." Bashirian et al. [27], working within the context of Protection Motivation Theory [28], found that fear was positively related to complying with COVID preventative behaviors among medical staff in Iran.

It is important to note that the present study concerned coping behaviors that were relatively easy to enact, hand washing, social distancing, and so forth. Many models of coping and reactions to stress suggest that the extent which people cope adaptively is positively related to whether people know what to do, whether they believe they can do what is adaptive, and whether they believe that these behaviors will be effective. In other words, people are more likely to cope when self-efficacy is high. See Schwarzer [29] for a discussion of the roles of self-efficacy in health behavior. Moreover, when self-efficacy is high (as was likely the case with the present coping behaviors), the motivation provided by fear can lead to more and more adaptive coping. A meta analysis of research on Rogers's Protection Motivation Theory also suggests the same conclusion [30].

In this regard, it is important to keep in mind that the present study was conducted at the beginning of the pandemic in Poland. There were active cases, the authorities had just begun to impose restrictions on people's movements, and at that time they strongly recommended what individuals should do to avoid infection with COVID-19, but the full effects of the pandemic on daily life had not occurred. This is why we emphasize the importance of the threat-emotion relationship in the beginning of the stress reaction process. People knew of the threats, and there were individual differences in how severe these threats were seen to be, and there were associated individual differences in the anxiety these threats created. It is possible that the relationships we found might have been different if we had conducted the study at a different point in the progression of the pandemic in Poland, but we were interested in examining relationships at the beginning of the pandemic.

Moreover, our results suggest that it is anxiety (or more generally speaking, negative activated affect) that motivates coping behavior, not general negative affect. We did not find that sadness/hopeless (or more generally speaking, negative deactivated affect) mediated relationships between threat and coping behavior. As suggested by the title, negative deactivated affect is not associated with engaging the environment; it is associated with withdrawing from the behavioral field. Hopelessness and powerlessness are not emotional reactions that spur people to take action.

## Practical implications

Although our study was conducted within the context of a pandemic of a specific virus in a specific place at a specific point in time, we think our results have implications for understanding how people react to and cope with problems that exist at a societal level. First, and most important, our results suggest that anxiety can serve a positive function. Much of the research on coping has focused on how people can reduce anxiety in the face of stress, and although reducing anxiety may be desirable in the long term, our results clearly indicate that feeling anxious was positively related to engaging in coping behaviors.

As suggested by Witte [9], it can be good to be afraid. Fear (and its affective partner, anxiety) can motivate people. The possibility that fear can be adaptive was also discussed by Harper et al. [25]. They noted that: "researchers and mental health professionals would be mindful to consider the context within which negative emotional states are experienced before considering whether such emotional states are necessarily pathological." Nevertheless, how this motivation is channeled is distinct from the importance of it as an initiator of a process. For example, climate change is a pressing societal problem, a catastrophe in waiting; yet, collective action to mitigate climate change, which is needed, is much less common than the worldwide collective action that has occurred in response to COVID-19. Perhaps if people were more afraid of the negative consequences of climate change, they (and society writ large) would take more action.

When discussing the positive role that fear may play in increasing compliance with COVID preventative behaviors, it is important to recognize that people can be "too afraid" to take action, a state that we defined as panic. Note that the item"paralyzed by fear" was part of this measure. We found that panic was positively related to self-preservation behaviors (e.g., stockpiling) and was negatively related to support for economic sacrifice, i.e., panic was related to self-interest not group-interest. Such relationships are exactly what Van Bavel et al. [6] described when discussing how extreme fear could undermine collective action. Moreover, in terms of research on fear appeals, it has long been recognized that fear appeals can be strong. It is adaptive to be afraid; it is maladaptive to be panicked.

Our results also highlight the potential importance of external events on coping behavior. Models of coping tend to focus on intrapsychic factors such as emotions or feelings of competence. Admittedly, external events exist as influences on coping behaviors only to the extent that they become internalized in some way. Nevertheless, the occurrence of the first coronavirus-related fatality in Poland "changed the playing field." Anxiety and coping behaviors both increased after this event.

Although this tragic event had desirable outcomes for Polish society as a whole such as greater behavioral engagement in the prevention of the spread of COVID-19, external events do not always have desirable consequences. For example, reports of individuals who claim to have been cured by some types of "natural" remedies (e.g., eating large quantities of garlic) may lead people to abandon the science-based recommendations of the authorities. Such possibilities seem to be more common given the easy access to false claims provided by the internet, and the authorities need to monitor and counteract such false claims [31,32].

## Limitations and conclusions

One of the shortcomings of static mediational models is that it is difficult to compare the explanatory power of different mediational paths. For example, we examined if emotions mediated relationships between threat and coping. In contrast, some research suggests (and has examined) the possibility that coping mediates relationships between threat and emotional reactions. Although we cannot compare the strength of these two mediational paths statistically, there are reasons to believe that, at least at the beginning of the pandemic, the path we proposed and examined is stronger than a path from threat to coping to emotions.

This support comes from comparisons of what is sometimes referred to as the "ab-path" in mediation. We compared the indirect effects of threat from the present analyses (threat to emotion to coping) to the indirect effects of threat from analyses that examined threat to coping to emotion mediation. In all cases, the indirect effects of the present model (threat through emotion to coping) were stronger than the indirect effects of threat through coping to emotion. These differences ranged from 2 to 5 standard errors. Although not definitive, such comparisons support our contention that the mediating role of emotional reactions in

relationships from threat to coping was stronger than the mediating role of emotional reactions in relationships from coping to threat.

Examining such possibilities requires collecting data over time, a design typically referred to as a panel design. Such data provide the opportunity to compare lagged relationships, e.g., the relationships between threat at time T1 and emotional reactions at time T2 and between emotional reactions at time T1 and threat at time T2. Such comparisons can provide insights into possible causal relationships between variables.

Karademas, Bati, Karkania, Georgiou, and Sofokleous [33] conducted a panel study about the H1N1 pandemic in 2009. They found that emotionality about H1N1 at the beginning of the pandemic (T1) was related to adaptive coping behavior four months later (T2). Unfortunately, Karademas et al., did not examine the reverse relationship, how coping at T1 was related to emotionality at T2.

Nevertheless, their results support our contention that emotional reactions to pandemics can motivate people to cope adaptively with a pandemic. More research is needed to understand the dynamics of these relationships. For example, are causal relationships between these constructs bi-directional, does the relative strength of these causal relationships change, and so forth?

By design, the present study examined responses to the COVID pandemic through the lens of a specific model, Witte's Extended Parallel Process Model [9]. This model primarily concerns the roles that fear can play in coping with distress. Nevertheless, positive emotions can also play important roles. For example, Heffner et al. [34] found that emotionally positive messages (prosocial messages) can increase the likelihood that people self-isolate. In addition, we focused on the roles played by affect and emotion. Clearly, more cognitively focused factors need to be taken into account. For example, a recent study found that a cognitively focused prime (highlighting the importance of relying on reason when making decisions) increased intentions to wear a mask, whereas an emotionally focused prime (highlighting the importance of relying on emotions when making decisions) did not affect intentions to wear a mask [35].

In terms of cognitive considerations, there is also ample evidence that people are either intentionally misinformed about or misinterpret recommendations for how to cope with COVID [31,32]. Such realities represent a challenge to scientists and public health practitioners. There have been discussions of what can be done to counteract misinformation [36], but the task is daunting.

The COVID-19 pandemic is an international event of historic proportions. To develop and implement effective counter-measures it is critical to understand how people react to and cope with this dangerous illness. As discussed by Van Bavel et al. [6], psychological science has much to offer in the fight against COVID. We hope that despite its limitations, the present paper contributes to this effort.

## Author Contributions

**Conceptualization:** Marzena Cypryańska.

**Formal analysis:** Marzena Cypryańska, John B. Nezlek.

**Methodology:** Marzena Cypryańska.

**Writing – original draft:** Marzena Cypryańska, John B. Nezlek.

**Writing – review & editing:** Marzena Cypryańska, John B. Nezlek.

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
