## [Decision Letter · Decision Letter 0]

2 Jul 2020

PONE-D-20-10016

Anxiety as a Mediator of Relationships between Perceptions of the Threat of COVID-19 and Coping Behaviors during the onset of the Pandemic in Poland

PLOS ONE

Dear Dr. Cypryanska,

Thank you for submitting your manuscript to PLOS ONE. After careful consideration, we feel that it has merit but does not fully meet PLOS ONE’s publication criteria as it currently stands. Therefore, we invite you to submit a revised version of the manuscript that addresses the points raised during the review process.

Please find below the reviewers' and mine's comments.

We look forward to receiving your revised manuscript.

Kind regards,

Valerio Capraro

Academic Editor

PLOS ONE

Journal Requirements:

2. Please provide additional details regarding participant consent. In the Methods section, please ensure that you have specified (1) whether consent was informed and (2) what type you obtained (for instance, written or verbal). If your study included minors, state whether you obtained consent from parents or guardians. If the need for consent was waived by the ethics committee, please include this information.

3. Thank you for your ethics statement:

'Komisja ds. Etyki Badań Naukowych

Uniwersytet SWPS, Filia w Poznaniu

Wydział Psychologii i Prawa

Approval number 2020-18-11

Data were analyzed anonymously.

Participants were free to terminate participation at any time.'

Please amend your current ethics statement to confirm that your named institutional review board or ethics committee specifically approved this study

Additional Editor Comments (if provided):

I have now collected two reviews from two experts in the field. The reviewers like the paper, but suggest several improvements. Therefore, I would like to invite you to revise your work following the reviewers' comments. Needless to say that all comments should be addressed. Moreover, while reading the manuscript, I have collected a few more comments that I think can improve the manuscript. They mainly regard the literature review, which seems to be rather short (only 20 references, many of which are not journal papers) and unfocused. I think this should definitely be improved. A good starting point could be the "perspective article" about what social and behavioural science can do to support Covid-19 response, that Van Bavel et al. 2020 have published in Nature Human Behaviour. Another suggestion is to look at the papers that have investigated messages and appeals to promote pandemic response (Bilancini et al. 2020; Capraro & Barcelo, 2020a; Capraro & Barcelo, 2020b; Everett et al. 2020; Heffner et al. 2020; Jordan et al. 2020). Of course, it is not a requirement to cite exactly these papers, but they might serve as a useful starting point to improve your literature review.

I am looking forward for the revision.

References

Bilancini E, Boncinelli L, Capraro V, Celadin T, Di Paolo R (2020) The effect of norm-based messages on reading and understanding COVID-19 pandemic response governmental rules. Journal of Behavioral Economics for Policy 4, 45-55.

Capraro, V., & Barcelo, H. (2020a). The effect of messaging and gender on intentions to wear a face covering to slow down COVID-19 transmission. arXiv preprint arXiv:2005.05467.

Capraro, V., & Barcelo, H. (2020b). Priming reasoning increases intentions to wear a face covering to slow down COVID-19 transmission. arXiv preprint arXiv:2006.11273.

Everett, J. A., Colombatto, C., Chituc, V., Brady, W. J., & Crockett, M. (2020). The effectiveness of moral messages on public health behavioral intentions during the COVID-19 pandemic. https://psyarxiv.com/9yqs8/

Heffner, J., Vives, M. L., & FeldmanHall, O. (2020). Emotional responses to prosocial messages increase willingness to self-isolate during the COVID-19 pandemic. https://psyarxiv.com/qkxvb/download?format=pdf

Jordan, J., Yoeli, E., & Rand, D. (2020). Don’t get it or don’t spread it? Comparing self-interested versus prosocially framed COVID-19 prevention messaging. https://psyarxiv.com/yuq7x

Van Bavel, J. J., et al. (2020). Using social and behavioural science to support COVID-19 pandemic response. Nature Human Behaviour.

Reviewers' comments:

Reviewer's Responses to Questions

**Comments to the Author**

1. Is the manuscript technically sound, and do the data support the conclusions?

Reviewer #1: Yes

Reviewer #2: Partly

2. Has the statistical analysis been performed appropriately and rigorously? 

Reviewer #1: Yes

Reviewer #2: Yes

3. Have the authors made all data underlying the findings in their manuscript fully available?

Reviewer #1: Yes

Reviewer #2: Yes

4. Is the manuscript presented in an intelligible fashion and written in standard English?

Reviewer #1: Yes

Reviewer #2: Yes

5. Review Comments to the Author

Reviewer #1: The MS presents results from a study that examined relationships between threat perceptions of, affective reactions and coping behaviors to Covid-19 at the start of the pandemic in Poland. Main results suggest that facets of emotion reactions to the pandemic (anxiety in particular) partially mediated relationships between aspects of Covid-19 threat perceptions (threat to self in particular) and coping behaviours. Moreover, although levels in all observed variables raised following the announcement of the first fatality that took place during conducting this study, this event did not seem influence the main relationships found which underlines the reliability of the tested model.

The MS and study have many things going for it. It utilizes a nationally representative study at the start of the pandemic, it distinguishes between self and collective attributions to threat perception, it distinguishes between different levels of negative emotion/affect and also measures some effective Covid-19 behaviors.

Notably, the main aimed contribution of this study is to revert relationships between threat perceptions, coping behavior and negative emotion, claiming that (negative) emotion in this case should be considered proximal to threat perception leading to respective coping behavior. The argument reads interesting and plausible. However, as a reader one would like some additional information about research (health, attitudinal or other) that went the other inverse way from that hypothesized in this study (i.e. from coping behaviors to affective reactions). In that way, the gravity of the argument will become clearer.

Authors need to substantiate analytically or otherwise their choice to include the two additional emotional states panicked, and paralyzed by fear.

Is there any evidence from data reduction to support the identification of the three negative emotion dimensions?

Discussion: I would expect that findings would help Social researchers better comprehend psychological processes involved in reaction to the pandemic; for example relationships found between threat to the self and affective reactions could inform research on self vs. social schemas (e.g., attachment or cultural orientations) regarding the pandemic. Other aspects of the study are also noteworthy supporting existing models in health behavior and extending those to pandemic-related behavior.

Overall, in discussing the main study concepts and their relationships, an effort could be made to ground those to the particular context, the start of the Covid-19 epidemic. Moreover, any information regarding likely similarities or differences with other contexts could be elaborated.

Minor points

Threat is 'perceived threat'

Emotions would be better described as 'Emotion' or 'Affective reactions to' Moreover it should be indicated as negative emotion or negative affective states

Access to the osf files can be made open to all in line with journal policy

Reviewer #2: PLOS One

MS# PONE-D-20-10016

Anxiety as a Mediator of Relationships between Perceptions of the Threat of COVID-19 and Coping Behaviors during the onset of the Pandemic in Poland

Comment to the Author(s):

The primary aim of the current study was to determine if anxiety, panic, and hopelessness mediated the relationship between perceived threat from the pandemic and various coping behaviors such as following WHO guidelines. We think these data are especially interesting because they were collected from a nationally representative sample at the beginning of the pandemic in Poland as individuals were just realizing the threat of the COVID-19 pandemic. However, we have several concerns about multiple aspects of the paper that would need to be made to the paper before we can recommend publication.

Concerns with the Introduction:

The Introduction, as written, is not as compellingly written as it could be. Primarily, we think the authors should better capitalize on the fact that these data were collected at the beginning of the pandemic. We agree that studying differences in how threat elicits emotion is important, especially during a time that could be confusing, shocking, and maybe even ambiguous for many people. Certainly, we know how the story turns out (as we write this, over 10 million cases worldwide and half a million deaths) but these data represent an window into how people pre-emptively think/feel/behave when stress is on the horizon. It is clear that the authors are aware of this strength as they write plainly in the Discussion, “this is why we emphasize the importance of the threat-emotion relationship in the beginning of the stress-reaction process”. However, this could be better introduced and expounded on in the Introduction. A smaller point, the authors should consider removing the research on efficacy from the Introduction. As written, it seems like this will be a part of the methodology. It is not (which is fine) but seems a bit of distraction.

Concerns with the Method/Measures:

Overall, this section of the paper was well-written and clear. A few issues of note:

• This paper would benefit from the addition of a procedure section.

o How were participants recruited? What kind of study did they believe they were going to take part in?

o Were the measures presented in a particular order? Counterbalanced?

o Were participants compensated in any way?

o It was unclear the order that the measures were presented in, and it was also not clear what participants were told about the study prior to signing up for the study.

• A larger concern with the methodology is the conceptualization of the economic sacrifice variable as a coping behavior. While spread prevention and self-preservation items assessed the extent to which participants took a number of specific actions, the authors used economic sacrifice items to ask about participant willingness to support certain policies or behaviors. This seems very different. First, the measurement of this variable is hypothetical in a way that self-preservation and spread prevention were not. Second, we are not sure that being willing to do something would result in stress, anxiety, or uncertainty reduction, which is what coping behaviors are hypothesized to do. We would recommend that the authors either remove this variable (we don’t think it is very compelling as measured) or make a stronger case for its conceptualization as a coping variable. (For example, can the authors show that this variable is significantly and positively related to both spread prevention and self-preservation?)

Concerns with the Results:

We had several concerns with the Results section. A word of praise first– the section was well- written and easy to follow.

• We were unable to judge the extent to which many variables were related to one another and therefore whether all analyses were actually warranted.

o Are panic and anxiety separate constructs? From a clinical perspective, they may not be (e.g., panic disorder and panic attacks are characterized as Anxiety Disorders in both the DSM-IV and the ICD-10. It could be that these two variables are highly correlated and would be better analyzed together in order to reduce the number of analyses conducted.

o We had the same question when considering the analyses related to threat (threat to self, Poland, and world) separately. The authors should give theoretical justification for keeping them separate or demonstrate statistically that they are not the same construct. If they are taping a similar construct, this too could reduce the number of necessary analyses.

• Second, the results seemed lengthier than they needed to be. Although we agree that it is important to discuss the a-to-c pathway (e.g. the relationship between threat and coping behaviors) before conducting mediational analyses, lengthy discussion of the other pathways is not necessary. We say this in part because the authors made it clear in the Introduction that a meditated relationship was always the intended prediction. Moving directly to the mediation analysis after establishing the a-c pathway is more efficient and in keeping with the theme of the paper.

• In addition, we did not feel that the analyses concerning differences in all variables before and after the first fatality were necessary and seemed a bit distracting. We say this in part because since these results did not moderate any relationships between variables and because there was no way in knowing whether or not participants knew about the first fatality in Poland. We think it was important that the authors assessed this; however, discussion of these results might be better left in a footnote or supplemental materials so as to not distract from the main purpose of the paper.

• Finally, we had some concerns about the factor analysis of the coping behaviors. It was unclear why all the items for self-preservation loaded with a negative value while the items for spread prevention loaded positively. The negative values might imply a need for reverse coding for the self-preservation items but based on the wording of the items, this doesn’t seem correct. A double check of the analyses and/or some clarification would be helpful.

Concerns with Discussion:

The Discussion section seemed quite divorced from the other parts of the paper. A few things of note:

• The authors seem to justify their results with theory that was not completely established in the Introduction. For instance, the authors discuss the Stress, Appraisal, and Coping model as a primary explanation for the results but this model was only mentioned briefly in the literature review. Further the discussion of feedback loops and bidirectional effects actually seem to weaken the results since those sorts of relationships were not (and could not be) tested.

• In other places, the Discussion seemed to lack connection to points made earlier in the paper. For example, the discussion of activated and deactivated affect (p. 20) could link back to the discussion of emotion as motivation in the Introduction to drive this point home, but it doesn’t.

• There were sections that did not seem to clearly make a point. For example, we were unclear as to the purpose of the paragraph on page 21 that discussed “desirable outcomes for Poland” and individuals’ interest in natural remedies. We also found the limitations and conclusion cursory. There was not a satisfactory discussion of the findings in context of caveats and the field more broadly. In addition, the paper just seems to end.

In sum, we think the authors have an interesting idea and some very valuable data. We hope our comments will be helpful to them and we wish them well with this work.

6. PLOS authors have the option to publish the peer review history of their article (what does this mean?). If published, this will include your full peer review and any attached files.

Reviewer #1: No

Reviewer #2: No

---

## [Author Response · Author response to Decision Letter 0]

12 Sep 2020

Response to reviewers

PONE-D-20-10016

Anxiety as a Mediator of Relationships between Perceptions of the Threat of COVID-19 and Coping Behaviors during the onset of the Pandemic in Poland

2. Please provide additional details regarding participant consent. In the Methods section, please ensure that you have specified (1) whether consent was informed and (2) what type you obtained (for instance, written or verbal). If your study included minors, state whether you obtained consent from parents or guardians. If the need for consent was waived by the ethics committee, please include this information.

Reply: All RODO and consent was managed by Ariadna Research Panel, the survey company (https://panelariadna.pl/)

3. Thank you for your ethics statement:

'Komisja ds. Etyki Badań Naukowych

Uniwersytet SWPS, Filia w Poznaniu

Wydział Psychologii i Prawa

Approval number 2020-18-11

Data were analyzed anonymously.

Participants were free to terminate participation at any time.'

Please amend your current ethics statement to confirm that your named institutional review board or ethics committee specifically approved this study. Once you have amended this/these statement(s) in the Methods section of the manuscript, please add the same text to the “Ethics Statement” field of the submission form (via “Edit Submission”). For additional information about PLOS ONE ethical requirements for human subjects research, please refer to http://journals.plos.org/plosone/s/submission-guidelines#loc-human-subjects-research.

Reply: We provided the required changes. 

Additional Editor Comments (if provided):

I have now collected two reviews from two experts in the field. The reviewers like the paper, but suggest several improvements. Therefore, I would like to invite you to revise your work following the reviewers' comments. Needless to say that all comments should be addressed. Moreover, while reading the manuscript, I have collected a few more comments that I think can improve the manuscript. They mainly regard the literature review, which seems to be rather short (only 20 references, many of which are not journal papers) and unfocused. I think this should definitely be improved. A good starting point could be the "perspective article" about what social and behavioural science can do to support Covid-19 response, that Van Bavel et al. 2020 have published in Nature Human Behaviour. Another suggestion is to look at the papers that have investigated messages and appeals to promote pandemic response (Bilancini et al. 2020; Capraro & Barcelo, 2020a; Capraro & Barcelo, 2020b; Everett et al. 2020; Heffner et al. 2020; Jordan et al. 2020). Of course, it is not a requirement to cite exactly these papers, but they might serve as a useful starting point to improve your literature review.

I am looking forward for the revision.

References

Bilancini E, Boncinelli L, Capraro V, Celadin T, Di Paolo R (2020) The effect of norm-based messages on reading and understanding COVID-19 pandemic response governmental rules. Journal of Behavioral Economics for Policy 4, 45-55. Posted: May 2020

Capraro, V., & Barcelo, H. (2020a). The effect of messaging and gender on intentions to wear a face covering to slow down COVID-19 transmission. arXiv preprint arXiv:2005.05467. posted: May 11, 2020

Capraro, V., & Barcelo, H. (2020b). Priming reasoning increases intentions to wear a face covering to slow down COVID-19 transmission. arXiv preprint arXiv:2006.11273. posted: June 16, 2020

Everett, J. A., Colombatto, C., Chituc, V., Brady, W. J., & Crockett, M. (2020). The effectiveness of moral messages on public health behavioral intentions during the COVID-19 pandemic. https://psyarxiv.com/9yqs8/ posted: March 20, 2020

Heffner, J., Vives, M. L., & FeldmanHall, O. (2020). Emotional responses to prosocial messages increase willingness to self-isolate during the COVID-19 pandemic. https://psyarxiv.com/qkxvb/download?format=pdf posted: April 15, 2020

Jordan, J., Yoeli, E., & Rand, D. (2020). Don’t get it or don’t spread it? Comparing self-interested versus prosocially framed COVID-19 prevention messaging. https://psyarxiv.com/yuq7x posted: April, 3, 2020

Van Bavel, J. J., et al. (2020). Using social and behavioural science to support COVID-19 pandemic response. Nature Human Behaviour. My draft manuscript copy is dated April 11. I am not certain when the original draft was first released.

Reply: 

We agree that it is best for studies to be placed in an informative and appropriate context, and when doing so, it is important to be transparent about what motivated a study. Why was it done? What factors were taken into consideration? In the introduction to our paper, we discussed the fact that our study was designed using previous research on pandemics as a guide and framework. The primary model was Witte’s model of the role of fear as a motivator. The study was not designed to provide an all-purpose, broad, explanation of everything that might influence or be related to responses to a pandemic. It was meant to be a focused study on how fear might serve as a positive motivator for coping with the pandemic. We should note that no one suggested that we include research about pandemics that had been done before we conducted our study.

Nevertheless, we agree that it is appropriate and important to locate our study within the broader context of research on the COVID pandemic, and we have added a meaningful amount of material to the discussion that includes research that was done after we conducted our study and submitted our paper. Noting this, selecting citations for this expanded coverage was a bit arbitrary. For example, Research Gate recently posted that its COVID list had 96,000 papers. Assuming that only a small fraction (5%) of these concerned social scientific topics, this still leaves close to 5,000 papers that could be relevant to our concerns. Clearly, this is much too large a body of research to summarize meaningfully. We have made a good faith effort to highlight relevant findings.

In terms of why we included this material in the discussion rather than in the introduction, it is important to note that all of the suggested citations were published or posted after we designed our study, which was in late February. Moreover, our paper was submitted at the early stage of the explosion of research on the pandemic, and most the suggested papers were published or posted after we submitted our paper in early April. Introductions are meant to include work that influenced the design and execution of a study. Given this, we do not think that it is appropriate for us to describe how work that did not exist when we designed our study and did not exist when we wrote our paper influenced our study and the paper.

For example, if we discussed research on pro-sociality in the introduction, readers would (logically) want to know why we did not include something about pro-sociality in our study. If this topic was important enough to describe as a context, readers will be puzzled as to why it was not important enough to merit inclusion in the study per se. Given this, we have made only minor changes to the introduction. We included a citation to the Van Bavel paper because it reviewed some of the same literature we did but nothing more. To cite papers that did not exist before we designed and conducted our study would be disingenuous, if not unethical. 

Reviewer #1: 

The MS presents results from a study that examined relationships between threat perceptions of, affective reactions and coping behaviors to Covid-19 at the start of the pandemic in Poland. Main results suggest that facets of emotion reactions to the pandemic (anxiety in particular) partially mediated relationships between aspects of Covid-19 threat perceptions (threat to self in particular) and coping behaviours. Moreover, although levels in all observed variables raised following the announcement of the first fatality that took place during conducting this study, this event did not seem influence the main relationships found which underlines the reliability of the tested model.

The MS and study have many things going for it. It utilizes a nationally representative study at the start of the pandemic, it distinguishes between self and collective attributions to threat perception, it distinguishes between different levels of negative emotion/affect and also measures some effective Covid-19 behaviors.

Notably, the main aimed contribution of this study is to revert relationships between threat perceptions, coping behavior and negative emotion, claiming that (negative) emotion in this case should be considered proximal to threat perception leading to respective coping behavior. The argument reads interesting and plausible. However, as a reader one would like some additional information about research (health, attitudinal or other) that went the other inverse way from that hypothesized in this study (i.e. from coping behaviors to affective reactions). In that way, the gravity of the argument will become clearer.

Reply: We are not exactly certain what the word “revert” means in the present context. It appears that the reviewer is asking us to present/discuss research that does not support our hypotheses in the introduction. This is a bit unusual. Unfortunately, our cross-sectional design does not provide a basis to support causal relationships. Nevertheless, this is an important point, and we address this issue in the revised discussion.

Authors need to substantiate analytically or otherwise their choice to include the two additional emotional states panicked, and paralyzed by fear.

Reply: See below. We have also added further justification for distinguishing anxiety and panic, and we included this topic in the revised discussion.

Is there any evidence from data reduction to support the identification of the three negative emotion dimensions?

Reply: Although we recognize this reviewer’s concerns (and the concerns of the individuals who provided Rev 2), we think that the differences in the relationships we found as a function of distinguishing the three types of affect/emotions constitute a prima facie case for the value of making the distinctions we made. This paper was not intended to provide a basis to examine the factorial structure of emotional reactions to COVID. We would have needed to collect more data than was practical given the limits of the type of national survey we conducted. For example, we would have probably had at least 4, probably 5 items for panic as we did for the other constructs. 

Discussion: I would expect that findings would help Social researchers better comprehend psychological processes involved in reaction to the pandemic; for example relationships found between threat to the self and affective reactions could inform research on self vs. social schemas (e.g., attachment or cultural orientations) regarding the pandemic. Other aspects of the study are also noteworthy supporting existing models in health behavior and extending those to pandemic-related behavior.

Overall, in discussing the main study concepts and their relationships, an effort could be made to ground those to the particular context, the start of the Covid-19 epidemic. Moreover, any information regarding likely similarities or differences with other contexts could be elaborated.

Reply: We appreciate the reviewer’s suggestions for broadening the discussion of our results, and we have added some material to the discussion to include such material.

Minor points

Threat is 'perceived threat'

Reply: We discuss perceived threat throughout the manuscript (it’s in the title of the paper), but we have added “perceived” in various instances to ensure that readers understand this.

Emotions would be better described as 'Emotion' or 'Affective reactions to' Moreover it should be indicated as negative emotion or negative affective states.

Reply: Throughout the paper, we discuss only negative emotions/affective states. Moreover, we are not sure why it is important to use the singular “emotion” instead of the plural “emotions.” We are happy to change this if it is important.

Reviewer #2: 

Comment to the Author(s):

The primary aim of the current study was to determine if anxiety, panic, and hopelessness mediated the relationship between perceived threat from the pandemic and various coping behaviors such as following WHO guidelines. We think these data are especially interesting because they were collected from a nationally representative sample at the beginning of the pandemic in Poland as individuals were just realizing the threat of the COVID-19 pandemic. However, we have several concerns about multiple aspects of the paper that would need to be made to the paper before we can recommend publication.

Concerns with the Introduction:

The Introduction, as written, is not as compellingly written as it could be. Primarily, we think the authors should better capitalize on the fact that these data were collected at the beginning of the pandemic. We agree that studying differences in how threat elicits emotion is important, especially during a time that could be confusing, shocking, and maybe even ambiguous for many people. Certainly, we know how the story turns out (as we write this, over 10 million cases worldwide and half a million deaths) but these data represent an window into how people pre-emptively think/feel/behave when stress is on the horizon. It is clear that the authors are aware of this strength as they write plainly in the Discussion, “this is why we emphasize the importance of the threat-emotion relationship in the beginning of the stress-reaction process”. However, this could be better introduced and expounded on in the Introduction. A smaller point, the authors should consider removing the research on efficacy from the Introduction. As written, it seems like this will be a part of the methodology. It is not (which is fine) but seems a bit of distraction.

Reply: We are not certain about how to capitalize any further on the timing of our study. To do so transparently, we would have to have some background research and theory to guide us. There is none. In terms of efficacy, we have expanded our discussion of this in the revised discussion section.

Concerns with the Method/Measures:

Overall, this section of the paper was well-written and clear. A few issues of note:

• This paper would benefit from the addition of a procedure section.

o How were participants recruited? What kind of study did they believe they were going to take part in?

o Were the measures presented in a particular order? Counterbalanced?

o Were participants compensated in any way?

o It was unclear the order that the measures were presented in, and it was also not clear what participants were told about the study prior to signing up for the study.

Reply: We did not include details about how participants were recruited because the data were collected by a professional company using the standard procedures used by such companies. We simply assumed that readers would be familiar with how survey companies collect data and so we do not include many details about this. For panels such as this, participants are contacted by the survey company with whom they have an agreement. They are invited to participate in a survey. It is not like undergraduates who sign up for a study. More details are available on the Ariadna website. Finally, we agree that it is worth noting the order in which the measures were presented, and we have done so. In addition, we modified the heading to include Procedures.

• A larger concern with the methodology is the conceptualization of the economic sacrifice variable as a coping behavior. While spread prevention and self-preservation items assessed the extent to which participants took a number of specific actions, the authors used economic sacrifice items to ask about participant willingness to support certain policies or behaviors. This seems very different. First, the measurement of this variable is hypothetical in a way that self-preservation and spread prevention were not. Second, we are not sure that being willing to do something would result in stress, anxiety, or uncertainty reduction, which is what coping behaviors are hypothesized to do. We would recommend that the authors either remove this variable (we don’t think it is very compelling as measured) or make a stronger case for its conceptualization as a coping variable. (For example, can the authors show that this variable is significantly and positively related to both spread prevention and self-preservation?)

Reply: We understand the reviewers’ concerns here. Inadvertently, for some results, we used labels of tables that did not always maintain the distinction between economic sacrifice and coping, although the text describing the results in these tables did not make the distinction. We have changed the table labels. We should also note that measures of coping were presented separately from measures of economic sacrifice, and measures of economic sacrifice were not included in the factor analyses of coping.

Concerns with the Results:

We had several concerns with the Results section. A word of praise first– the section was well- written and easy to follow.

• We were unable to judge the extent to which many variables were related to one another and therefore whether all analyses were actually warranted.

Reply: When analyzing correlated measures there is always a tension between combining measures and analyzing them separately. We chose to analyze some measures separately for various reasons. First, given the newness of the topic, we thought that examining measures separately would be informative. COVID present unique challenges and stresses, and if measures were combined prior to analysis, important differences would/could be lost. We think it useful to keep in mind that when two variables are correlated .7 (which many might consider as reasonable basis for combining the variables into a single measure) 50% of the variance is unshared. Moreover, as we discuss below, we did find differences among measures in all of the instances for which the reviewers suggested that it might be better to combine measures. We have six tables, and following the suggestions of the reviewers we might have five, or six with fewer entries.

o Are panic and anxiety separate constructs? From a clinical perspective, they may not be (e.g., panic disorder and panic attacks are characterized as Anxiety Disorders in both the DSM-IV and the ICD-10. It could be that these two variables are highly correlated and would be better analyzed together in order to reduce the number of analyses conducted.

Reply: Although we appreciate this point, we think distinguishing panic and anxiety is worthwhile. First, the fact that panic disorders are part of anxiety disorders does not mean that panic and anxiety cannot be separated. In fact, the existence of a separate diagnoses for panic disorders suggests there is something specific about panic vs. general anxiety when discussing disorders. Moreover, using the structure of clinical diagnoses as a template for measuring non-clinical emotional experience is fraught with problems.

Most important, in regression analyses in which anxiety and panic are entered as separate predictors, meaning that the covariance between the two was part of the model (e.g., Table 3), panic was related to outcomes above and beyond relationships between anxiety and the outcomes. In fact, for economic sacrifice, the coefficient for panic was negative, whereas the coefficient for anxiety was positive. See similar differences in Table 4. Such differences constitute a prima facie case for distinguishing these two measures.

Finally, we have included new citations supporting the importance of this distinction. 

o We had the same question when considering the analyses related to threat (threat to self, Poland, and world) separately. The authors should give theoretical justification for keeping them separate or demonstrate statistically that they are not the same construct. If they are taping a similar construct, this too could reduce the number of necessary analyses.

Reply: Although we appreciate the point the reviewers make, there are clear differences in the coefficients (Table 4) for the three types of identity. Moreover, because the three types of identity are entered simultaneously, the covariances between them are taken into account. We believe this qualifies as a “statistical demonstration” that it is worthwhile to distinguish the three threats. Simply because each is a measure threat is not a basis to assume that they have the same relationships with other measures (which they do not) and then combine them into a single measure.

• Second, the results seemed lengthier than they needed to be. Although we agree that it is important to discuss the a-to-c pathway (e.g. the relationship between threat and coping behaviors) before conducting mediational analyses, lengthy discussion of the other pathways is not necessary. We say this in part because the authors made it clear in the Introduction that a meditated relationship was always the intended prediction. Moving directly to the mediation analysis after establishing the a-c pathway is more efficient and in keeping with the theme of the paper.

Reply: This is simply a matter of style. We thought (and still think) that a thorough presentation of the results is appropriate given the subject matter and the hypotheses. Yes, the focus of the paper was on the mediational analyses, but we think that the other relationships merited presentation. Although presenting these analyses made the paper longer, it appears that this did not interfere with the reviewers’ comprehension, as judged by their own words: “the section was well- written and easy to follow.’

• In addition, we did not feel that the analyses concerning differences in all variables before and after the first fatality were necessary and seemed a bit distracting. We say this in part because since these results did not moderate any relationships between variables and because there was no way in knowing whether or not participants knew about the first fatality in Poland. We think it was important that the authors assessed this; however, discussion of these results might be better left in a footnote or supplemental materials so as to not distract from the main purpose of the paper.

Reply: With respect, we disagree. In terms of whether participants knew about the first fatality, we carefully classified participants in terms of when the information was made public (by the hour), and deleted participants who we could not be reasonably certain had heard the news. In Poland, this news “spread like wildfire.” We could document the numerous news outlets that carried the story, but we do not think there is a need. Admittedly, there is no control group, but we could not anticipate this would happen. Although the relationships between measures did not change after the first fatality, the means certainly did, and means matter. Our paper is about how people reacted to the COVID pandemic and news of death is part of the situation to which people are reacting. 

• Finally, we had some concerns about the factor analysis of the coping behaviors. It was unclear why all the items for self-preservation loaded with a negative value while the items for spread prevention loaded positively. The negative values might imply a need for reverse coding for the self-preservation items but based on the wording of the items, this doesn’t seem correct. A double check of the analyses and/or some clarification would be helpful.

Reply: We understand the reviewers’ confusion about this issue. We neglected to mention that the correlation between the two factors was negative, which is consistent with the negative loadings on the second factor. The composite scores (scale scores) were positively correlated. This is explained in the revised manuscript.

Concerns with Discussion:

The Discussion section seemed quite divorced from the other parts of the paper. A few things of note:

• The authors seem to justify their results with theory that was not completely established in the Introduction. For instance, the authors discuss the Stress, Appraisal, and Coping model as a primary explanation for the results but this model was only mentioned briefly in the literature review. Further the discussion of feedback loops and bidirectional effects actually seem to weaken the results since those sorts of relationships were not (and could not be) tested.

Reply: We are not certain what the term “completely established” means. In the introduction, we described the relevance of the Lazarus work to our project and provided citations to this. This description is a three-sentence paragraph. It is short, but our intention was to include and describe only these elements of the of Lazarus’s model that were directly relevant to our study. The next paragraph elaborates on this point and presents support for the mediational hypothesis.

Also, we do not think that a discussion of feedback loops and bidirectionality weakens our argument As we discuss, such processes are possible, but their existence does not preclude the existence of the processes we document. We do not believe that our study can address such issues, but as we discuss in the next paragraph, no previous study has examined the mediational relationships we did, relationships that are assumed to underlie Lazarus’s model.

• In other places, the Discussion seemed to lack connection to points made earlier in the paper. For example, the discussion of activated and deactivated affect (p. 20) could link back to the discussion of emotion as motivation in the Introduction to drive this point home, but it doesn’t.

Reply: The paper is about emotion as a motivator. Following the discussion of active and deactive affect, there is a paragraph that starts with the following: “Moreover, our results suggest that it is anxiety (or more generally speaking, negative activated affect) that motivates coping behavior, not general negative affect.” This seemed to be direct to us.

• There were sections that did not seem to clearly make a point. For example, we were unclear as to the purpose of the paragraph on page 21 that discussed “desirable outcomes for Poland” and individuals’ interest in natural remedies. We also found the limitations and conclusion cursory. There was not a satisfactory discussion of the findings in context of caveats and the field more broadly. In addition, the paper just seems to end.

Reply: It is a bit difficult to reply to this comment. In terms of the paragraph on p. 21 the reviewers mention, the purpose of the paragraph is stated in the first sentence: “Although this tragic event had desirable outcomes for Polish society as a whole such as greater behavioral engagement in the prevention of the spread of COVID-19, singular events may not always have desirable consequences.” The point is that although singular events can focus attention, they can also overpower science. We included reference to the “infodemic” that considers this issue. In terms of the limitations and conclusions, just before this was a section “Practical applications,” which we think considered some of these issues. Moreover, we expanded the limitations and conclusions section.

In sum, we think the authors have an interesting idea and some very valuable data. We hope our comments will be helpful to them and we wish them well with this work.

---

## [Decision Letter · Decision Letter 1]

29 Sep 2020

PONE-D-20-10016R1

Anxiety as a Mediator of Relationships between Perceptions of the Threat of COVID-19 and Coping Behaviors during the onset of the Pandemic in Poland

PLOS ONE

Dear Dr. Cypryanska,

Thank you for submitting your manuscript to PLOS ONE. After careful consideration, we feel that it has merit but does not fully meet PLOS ONE’s publication criteria as it currently stands. Therefore, we invite you to submit a revised version of the manuscript that addresses the points raised during the review process.

We look forward to receiving your revised manuscript.

Kind regards,

Valerio Capraro

Academic Editor

PLOS ONE

Additional Editor Comments (if provided):

One of the reviewer suggests some additional minor improvements before publication. Please address these last comments at your earliest convenience. I am looking forward for the final version.

Reviewers' comments:

Reviewer's Responses to Questions

**Comments to the Author**

1. If the authors have adequately addressed your comments raised in a previous round of review and you feel that this manuscript is now acceptable for publication, you may indicate that here to bypass the “Comments to the Author” section, enter your conflict of interest statement in the “Confidential to Editor” section, and submit your "Accept" recommendation.

Reviewer #1: (No Response)

Reviewer #2: All comments have been addressed

2. Is the manuscript technically sound, and do the data support the conclusions?

Reviewer #1: Yes

Reviewer #2: Yes

3. Has the statistical analysis been performed appropriately and rigorously? 

Reviewer #1: Yes

Reviewer #2: Yes

4. Have the authors made all data underlying the findings in their manuscript fully available?

Reviewer #1: Yes

Reviewer #2: Yes

5. Is the manuscript presented in an intelligible fashion and written in standard English?

Reviewer #1: Yes

Reviewer #2: Yes

6. Review Comments to the Author

Reviewer #1: Authors have addressed points raised in the first round of reviews and the revised MS is improved as a whole. Below a few, minor, points that can improve the final MS. In the opening paragraphs of the introduction the main focus of research could be better communicated. I am referring in particular in the logical progression from the second to the third paragraph.

For the sake of parsimony, at places you could specify that the research is on negative emotion/emotional states/emotional reactions and not emotion/s at large. This can actually improve argumentation at places (i.e., p. 4 ‘More specifically, we examined if emotions mediate relationships between perceived threat and coping behavior’ and p. 5 ‘in this paper, we focus on the mediating role of emotions’ and elsewhere).

Similarly, you could provide examples of supporting research to statements such as in p. 3 “Our focus on relationships between threat and coping and between threat and emotions was based on previous research that has consistently found that the perceived threat of an infectious disease is positively related to engaging in coping behaviors.”

The terms emotion, emotions, emotional reactions, affect, affective reactions etc. are used interchangeably throughout the MS. This could be flagged in a footnote. In the first round of reviews I suggested to reserve the plural form of the term (i.e., emotions) to categorical forms of emotional states. Generally, the singular form tends to be used to signify more broad emotion-related processes such as the ones examined in this study. Such a stance could also ease the issue of considering (or not) Hopelessness and powerlessness (p. 22) an ‘emotion’ (I would not consider those as emotions).

p. 6 "Responses to …. different perspectives". Unless you provide some examples of some of the perspectives referred to in this sentence, the second part of the sentence in particular, is quite vague in meaning.

p. 12 ‘the’ WHO

p. 21 meta analysis

Reviewer #2: (No Response)

7. PLOS authors have the option to publish the peer review history of their article (what does this mean?). If published, this will include your full peer review and any attached files.

Reviewer #1: No

Reviewer #2: No

---

## [Author Response · Author response to Decision Letter 1]

14 Oct 2020

Reviewer #1: 

Authors have addressed points raised in the first round of reviews and the revised MS is improved as a whole. Below a few, minor, points that can improve the final MS. In the opening paragraphs of the introduction the main focus of research could be better communicated. I am referring in particular in the logical progression from the second to the third paragraph.

Reply: Good point. We have revised the third paragraph to make this clearer.

For the sake of parsimony, at places you could specify that the research is on negative emotion/emotional states/emotional reactions and not emotion/s at large. This can actually improve argumentation at places (i.e., p. 4 ‘More specifically, we examined if emotions mediate relationships between perceived threat and coping behavior’ and p. 5 ‘in this paper, we focus on the mediating role of emotions’ and elsewhere).

Reply: We have added “negative” when appropriate in numerous instances, and as noted below, we have used the term “emotional reactions.” 

Similarly, you could provide examples of supporting research to statements such as in p. 3 “Our focus on relationships between threat and coping and between threat and emotions was based on previous research that has consistently found that the perceived threat of an infectious disease is positively related to engaging in coping behaviors.”

Reply: Good point. We added three citations for this specific point, and we included a reference in the discussion to research on self-efficacy in health behavior. 

The terms emotion, emotions, emotional reactions, affect, affective reactions etc. are used interchangeably throughout the MS. This could be flagged in a footnote. In the first round of reviews I suggested to reserve the plural form of the term (i.e., emotions) to categorical forms of emotional states. Generally, the singular form tends to be used to signify more broad emotion-related processes such as the ones examined in this study. Such a stance could also ease the issue of considering (or not) Hopelessness and powerlessness (p. 22) an ‘emotion’ (I would not consider those as emotions).

Reply: The reviewer makes a good point. We agree that it is important to be precise, and when referring to the present study, we have used the phrase “emotional reaction,” not “emotion,” throughout the paper. PLOS One does not permit footnotes, so we have inserted a brief sentence (top of p. 4, revision) describing the fact that our use of emotional reactions is broader than the construct of emotions as used in formal models of emotions. 

p. 6 "Responses to …. different perspectives". Unless you provide some examples of some of the perspectives referred to in this sentence, the second part of the sentence in particular, is quite vague in meaning.

Reply: We agree that this sentence is vague. It was intentionally so because as noted in the last sentence of this paragraph, we wanted to address the issue of alternative perspectives in the discussion, something we did. We did not think we needed to review in the introduction (even in passing), the sundry models that have been used to explain reactions to pandemics because our study was not designed to compare and evaluate these other approaches.

p. 12 ‘the’ WHO

Reply: In this instance, the article “the” is not appropriate because WHO is serving as an adjective.

p. 21 meta analysis

Reply: Done

---

## [Editor Report · Decision Letter 2]

16 Oct 2020

Anxiety as a Mediator of Relationships between Perceptions of the Threat of COVID-19 and Coping Behaviors during the onset of the Pandemic in Poland

PONE-D-20-10016R2

Dear Dr. Cypryanska,

We’re pleased to inform you that your manuscript has been judged scientifically suitable for publication and will be formally accepted for publication once it meets all outstanding technical requirements.

Kind regards,

Valerio Capraro

Academic Editor

PLOS ONE
---

## [Editor Report · Acceptance letter]

22 Oct 2020

PONE-D-20-10016R2 

Anxiety as a Mediator of Relationships between Perceptions of the Threat of COVID-19 and Coping Behaviors during the onset of the Pandemic in Poland 

Dear Dr. Cypryańska:

I'm pleased to inform you that your manuscript has been deemed suitable for publication in PLOS ONE. Congratulations! Your manuscript is now with our production department. 

Kind regards, 

on behalf of

Dr. Valerio Capraro 

Academic Editor

PLOS ONE